# AHash: A Load-Balanced One Permutation Hash

## Abstract

Minwise Hashing (MinHash) is a fundamental method to compute set similarities and compact high-dimensional data for efficient learning and searching. The bottleneck of MinHash is computing $k$ (usually hundreds) MinHash values. One Permutation Hashing (OPH) only requires one permutation (hash function) to get $k$ MinHash values by dividing elements into $k$ bins. One drawback of OPH is that the load of the bins (the number of elements in a bin) could be unbalanced, which leads to the existence of empty bins and false similarity computation. Several strategies for densification, that is, filling empty bins, have been proposed. However, the densification is just a remedial strategy and cannot eliminate the error incurred by the unbalanced load. Unlike the densification to fill the empty bins after they undesirably occur, our design goal is to balance the load so as to reduce the empty bins in advance. In this paper, we propose a load-balanced hashing, *Amortization Hashing (AHash)*, which can generate as few empty bins as possible. Therefore, AHash is more load-balanced and accurate without hurting runtime efficiency compared with OPH and densification strategies. Our experiments on real datasets validate the claim. All source codes and datasets have been released on GitHub anonymously [1].

## 1 Introduction

### 1.1 Background

MinHash (Broder et al., 2000; Broder, 1997) is a powerful tool in processing high-dimensional data (*e.g.*, texts) that is often viewed as a set. For a set, MinHash produces $k$ minimum hash values, which are called **MinHash values**. MinHash values can support similarity computing (Broder, 1997; Chien & Immorlica, 2005; Henzinger, 2006b; Bayardo et al., 2007), large-scale linear learning (Li & König, 2010; Li et al., 2011; 2012; Yu et al., 2012), fast near neighbour searching (Shrivastava & Li, 2014a; Li et al., 2010; Indyk & Motwani, 1998; Shrivastava & Li, 2012) and so on. Due to the importance of MinHash, many recent works strive to improve its performance such as One Permutation Hashing (OPH) (Li et al., 2012), Densified OPH (DOPH) (Shrivastava & Li, 2014a), and Optimal OPH (OOPH) (Shrivastava, 2017).

### 1.2 Prior Art and Their Limitations

**MinHash** (Broder et al., 2000; Broder, 1997) **is the most classical method but has unacceptable computation cost.** Given a set $S \in \mathscr{U}$ ($\mathscr{U}$ is the universe of all elements), MinHash applies $k$ random permutations (hash functions) $\pi_i : \mathscr{U} \to \mathscr{U}$, on all elements in $S$ (see Figure 1). For each hash function, it computes $|S|$ hash values but only maintains the minimum one. Then we get the $k$ *MinHash values*. In practice, to achieve high accuracy, users need to compute a large number of MinHash values, *i.e.*, $k$ needs to be very large. Unfortunately, computing $k$ hash functions for each element is a computational and resource bottleneck when $k$ is large. It is showed that some large-scale learning applications need to

---

[1]https://github.com/AHashCodes/AHash

compute more than 4000 MinHash values, which takes a non-negligible portion of the total computation time (Fernandez et al., 2019; Li, 2015).

**One Permutation Hash (OPH)** (Li et al., 2012) **overcomes the drawback of Min-Hash, but the produced MinHash values may be less than the demand.** To overcome the drawback of MinHash, OPH reduces the number of hash computations per element from $k$ to 1. As shown in Figure 1, the key idea of OPH is that all elements are divided into $k$ bins by hashing, and each bin maintains the minimum hash value respectively. Ideally, each bin will produce a MinHash value. Unfortunately, if the data is skewed or sparse, some bins could have many elements, but some bins could be empty. As a result, the number of produced MinHash values could be smaller than $k$.

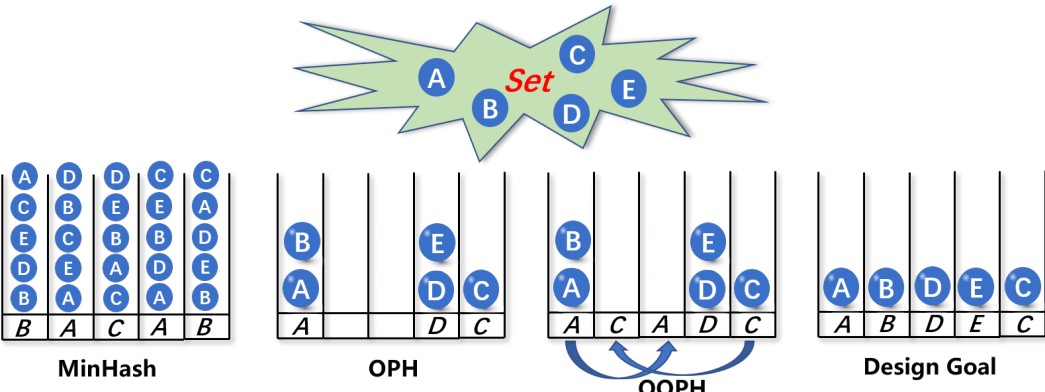

Figure 1: A toy example of existing works and our design goal with $k = 5$. The hash values of elements at the bottom are MinHash values.

**Existence of empty bins is fatal for some applications.** To compute similarities, it is known that the existence of empty bins leads to the false similarity estimation (see detailed reasons in Section 2) (Shrivastava, 2017; Shrivastava & Li, 2014a; Fernandez et al., 2019). To search near neighbours, using the empty bins as indexes leads to the false positives (see detailed reasons in Section 2) (Shrivastava & Li, 2012; 2014a). Therefore, every empty bin needs a MinHash value, and how to appropriately fill empty bins attracts research interests in recent years (Li et al., 2012; Shrivastava, 2017; Shrivastava & Li, 2014a;b).

**Densification is helpful for filling empty bins but does not fairly use the original data.** Densification, proposed by Shrivastava & Li (2014a), is to fill the empty bins by reusing the minimum values of the non-empty bins to fix the drawback of OPH. Many densification strategies have been proposed (Li et al., 2012; Shrivastava & Li, 2014a;b; Shrivastava, 2017). Among all densification strategies, OOPH (Shrivastava, 2017) achieves the smallest variance. It hashes IDs of empty bins to select a non-empty bin to reuse (See Figure 1). Although it achieves the smallest variance, it is often impossible to choose the optimal value to fill each empty bin. As shown as Figure 1, OOPH just reuses the MinHash values of non-empty bins. Because OPH only involves three elements $\{A, D, C\}$ to compute MinHash values, it places the constrains on OOPH which can only reuse these three elements. Such a reuse is unfair: the Minhash values could be reused multiple times, but other elements ($\{B, E\}$) will never be reused. Unfairness probably incurs error.

### 1.3 OUR SOLUTION: AMORTIZATION HASHING

Instead of filling empty bins, this paper proposes, Amortization Hashing (AHash), which can produce fewer empty bins, so as to achieve high accuracy. As shown in Figure 1, to minimize the unfairness of the densification, we aim to design a load-balanced hash (AHash) which produces as few empty bins as possible. After using Amortization Hashing, there could be still empty bins for very sparse sets, and we use OOPH to fill the remaining empty bins. We have proved (see Theorem 3) that the lower the number of empty bins is, the higher accuracy will be. This paper is the first attempt to design a load-balanced hash for OPH.

Intuitively, if a bin holds too many elements, it is **wasted**. In contrast, if a bin holds no element, it is **starved**. Therefore, if we can pair up a wasted bin and a starved bin, and amortize the elements between them, we can eliminate the unfairness of using densification. The details of amortization hashing are provided in Section 3.

AHash is a novel solution which is orthogonal with densification, and can improve accuracy of all densification strategies. For densification strategies, filling each empty bin could incur small error. Filling more empty bins will incur larger error. To minimize the error of filling empty bins, we are expected to minimize the number of empty bins during dividing elements into bins. It is worth mentioning that AHash does not incur additional computation cost.

### 1.4 Key Contributions

1) We propose a load-balanced hashing, Amortization Hashing, which can improve accuracy while retaining the same runtime efficiency. (Section 3)

2) We conduct theoretical analysis and extensive experiments, and results show that AHash can achieve higher accuracy. (Section 4)

3) We apply AHash for two data mining tasks, linear SVM, and near neighbour search, and results show AHash significantly outperforms the state-of-the-art in practical applications. (Section 4)

## 2 Preliminaries

### 2.1 Formal Defnitions of MinHash, OPH and OOPH

Formal definitions of MinHash, OPH and OOPH are as follows:

- MinHash (Broder, 1997; Broder et al., 2000): the $i^{th}$ MinHash value of $S$ is defined as:

$$h_i^{MinHash}(S) = \min_{0 \leq i < k}\{\pi_i(S)\} \tag{1}$$

  where $\pi_i(S)$ denotes the $i^{th}$ hashing on $S$.

- One Permutation Hashing (Li et al., 2012): the $i^{th}$ MinHash value of $S$ is defined as:

$$h_i^{OPH}(S) = \begin{cases} \min\{\pi(S) \cap \Omega_i\} & \text{if } \{\pi(S) \cap \Omega_i \neq \emptyset\} \\ Empty & \text{no element falls in this bin} \end{cases} \tag{2}$$

  where $\pi(S)$ denotes the hashing on $S$ and $\Omega_i$ denotes $i^{th}$ partition of the rang space of $\pi(S)$.

- One Permutation Hashing with Optimal Densification (Shrivastava, 2017): the $i^{th}$ MinHash value of $S$ is defined as:

$$h_i^{OOPH}(S) = \begin{cases} \min\{\pi(S) \cap \Omega_i\} & \text{if } \{\pi(S) \cap \Omega_i \neq \emptyset\} \\ h_j^{OOPH}(S) & \text{no element falls in this bin} \end{cases} \tag{3}$$

  where $h_{univ}(i)$ denotes 2-universal hashing (Shrivastava, 2017) and $j = h_{univ}(i)$.

### 2.2 Use MinHash Values for Computing Similarities

Computing similarities is a key step for many applications like duplicate detection (Broder, 1997; Henzinger, 2006a), semantic similarity estimation (Chien & Immorlica, 2005), frequent pattern mining (Buehrer & Chellapilla, 2008; Chierichetti et al., 2009) and more. In the applications, the data (*e.g.*, texts) can be viewed as 0/1 binary data (*e.g.*, the absence/presence of a word), which is equivalent to a set. Given two sets, $S_1$ and $S_2$, the similarity is usually measured by *Jaccard Similarity*, which is defined as $J(S_1, S_2) = |S_1 \cap S_2|/|S_1 \cup S_2|$. For $S_1$ and $S_2$ with the same hashing, the probability of the two minimum hash values being the same is equal to *Jaccard Similarity* of $S_1$ and $S_2$, which is formally shown as follow:

$$\Pr[h(S_1) = h(S_2)] = \frac{|S_1 \cap S_2|}{|S_1 \cup S_2|} = J(S_1, S_2) \tag{4}$$

Such a property is called *Locality Sensitive Hash (LSH) Property* (Indyk & Motwani, 1998; Charikar, 2002). Note that, if $h(S_1)$ and $h(S_2)$ are simultaneously empty, they are certain to collide, which violates the LSH Property. So we must handle empty bins. Given MinHash values of $S_1$ and $S_2$, the $J(S_1, S_2)$ can be approximated as:

$$\hat{J}(S_1, S_2) = \frac{1}{k} \sum_{j=1}^{k} 1\{h_j(S_1) = h_j(S_2)\} \tag{5}$$

with $1\{x\}$ being the indicator function that takes value 1 when $x$ is true and otherwise 0.

### 2.3 Use MinHash Values for Large-scale Learning

Large-scale linear learning like training SVM (Fan et al., 2008; Hsieh et al., 2008; Joachims, 2006) is faced with extremely high-dimensional data, which emphasizes the application of hashing algorithms. Given a dataset $\{\boldsymbol{x_i}, y_i\}_{i=1}^{l}, \boldsymbol{x_i} \in R^n, y_i \in \{-1, +1\}$, L2-SVM solves the following unconstrained optimization problem:

$$\min_{\boldsymbol{w}} \quad \frac{1}{2}\boldsymbol{w}^T\boldsymbol{w} + C \sum_{i=1}^{l} max(1 - y_i\boldsymbol{w}^T\boldsymbol{x_i}, 0)^2 \tag{6}$$

where $C > 0$ is a penalty parameter.

MinHash can be used to compact the feature vectors to reduce the feature dimensionality if the dataset is binary (*i.e.*, each feature vector consists of only 0s and 1s) (Li et al., 2011). Actually, each feature vector can be viewed as a set: if the $i$-th element of the vector is 1, then the $i$-th element is in the set. Thus we can use MinHash values to represent a feature vector. We use one-hot encoding (Coates & Ng, 2011; Buckman et al., 2018) for each hash value and concatenate these $k$ values to get the new feature vector. To further reduce the dimension, Li et al. (2010) proposes that we can use only the lowest $b$ (*e.g.*, $b$=8) bits of each hash value (usually 64 bits). Thus the new feature vector is only $2^b \times k$-bit long, regardless of the dimensionality of the original data.

### 2.4 Use MinHash Values for Fast Near Neighbour Search

Fast near neighbour search is important in many areas like databases (Broder, 1997; Friedman et al., 1975) and machine learning (Shrivastava & Li, 2012), especially applications with high-dimensional data. Given a query set $S$, near neighbour search is to return other sets whose similarities with $S$ are more than a threshold.

Shrivastava & Li (2012) proposed MinHash values can be used in near neighbour search in sub-linear time complexity (*i.e.*, without scanning the whole dataset). Specifically, a *signature* for a set is generated by concatenating $k$ MinHash values like this:

$$Signature(S) = [h_1(S); h_2(S); ....; h_k(S)] \tag{7}$$

In this way, those similar sets are more likely to have the same *signature* (It is not true if empty bins occur). So we can build hash tables by using the signatures as indexes and the sets as the values of the hash table entries. Moreover, to reduce the number of hash table entries, we only concatenate the lowest $b$ bits of $k$ hashed values to generate signatures (Li et al., 2010). To improve the recall of the query results, we can calculate $L$ different signatures for each set and build $L$ hash tables, and return the union of entries from these $L$ hash tables as the query result. For different hash tables, we should use independent hash functions to compute MinHash values. Parameters $b$, $L$, and $k$ can be used to control the threshold of near neighbor search. Using $L$ hash tables, the processing cost of MinHash for one query set is $O(nkL)$ ($n$ is the size of the query set). Shrivastava & Li (2014a) proposes that OPH can generate $k \times L$ MinHash values only by one hashing with $k \times L$ bins, so the processing cost of OPH is $O(n + kL)$. ($O(n)$ is for hashing on $n$ elements and $O(kL)$ is for filling empty bins).

## 3 Algorithm

### 3.1 Amortization Hashing

---

**Algorithm 1:** Insertion

**Input:** $k$ bins $B[.]$, a hash function $h(.)$, a set $S$

1   $\Omega \leftarrow$ output range of $h(.)$
2   Initialize B[.].EvenMin=$+\infty$
    B[.].OddMin=$+\infty$
3   **for** *each element $e$ in $S$* **do**
4     $V \leftarrow h(e)$
5     $i \leftarrow h(e)/(\Omega/k)$
6     **if** *V%2=1* **then**
7       B[i].OddMin = min($V$, B[i].OddMin)
8     **if** *V%2=0* **then**
9       B[i].EvenMin = min($V$, B[i].EvenMin)

---

**Algorithm 2:** Amortization

**Input:** $k$ bins $B[.]$ after *Insertion*

1   **for** $i=0$*; $i < k$; $i+=2$* **do**
2     **if** *B[i] is empty and B[i+1] is non-empty* **then**
3       B[i].OddMin = B[i+1].EvenMin
4     **if** *B[i] is non-empty and B[i+1] is empty* **then**
5       B[i+1].EvenMin = B[i].OddMin
6   *If empty bins still exist:*
7     *Densification*

---

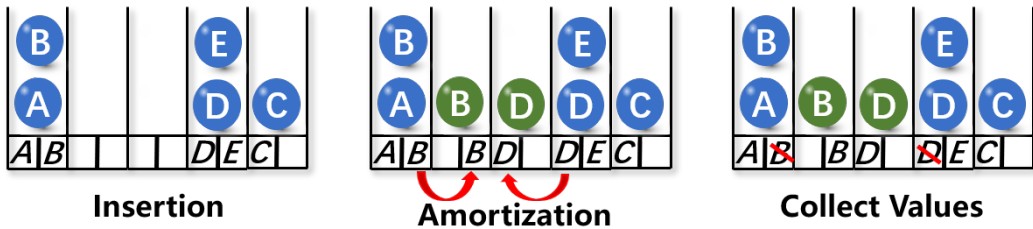

Figure 2: A toy example of AHash with $k = 5$.

To generate $k$ MinHash values, Amortization Hashing (AHash) has two key steps: *Insertion* and *Amortization*.

**Insertion**: As shown in Algorithm 1 and Figure 2, AHash divides all elements into $k$ bins by hashing and each bin maintains one **even min** and one **odd min**. Even min refers to the minimum one among all even hash values. Odd min refers to the minimum one among all odd hash values.

**Amortization**: As shown in Algorithm 2 and Figure 2, AHash pairs up $k$ bins as follows: $i^{th}$ and $(i+1)^{th}$ bins are a pair ($i = 0, 2, .., \lfloor \frac{k-2}{2} \rfloor$). For a pair, the $i^{th}$ bin is called **even bin** and the $(i+1)^{th}$ bin is called **odd bin**. For the pairs with only one empty bin, there are two cases:

- If the even bin $B_e$ is empty, we reassign $B_e.EvenMin$ with the even min of the odd bin.

- If the odd bin $B_o$ is empty, we reassign $B_o.OddMin$ with the odd min of the even bin.

For the pairs with two simultaneously non-empty or empty bins, we keep them unchanged. After the amortization, we collect $k$ MinHash values from $k$ bins and there are two cases (See Figure 2):

- For each even bin with both even and odd mins, we delete the odd min.

- For each odd bin with both even and odd mins, we delete the even min.

If there are still empty bins, we conduct densifying like OOPH, which rarely happens because AHash significantly reduces the empty bins. Formally, for even bins,

$$h_i^{AHash}(S) = \begin{cases} min\{\pi_e(S) \cap \Omega_i\} & \pi_e(S) \cap \Omega_i \neq \emptyset \\ min\{\pi_o(S) \cap \Omega_i\} & \pi_e(S) \cap \Omega_i = \emptyset \ and \ \pi_o(S) \cap \Omega_i \neq \emptyset \\ min\{\pi_e(S) \cap \Omega_j\} & i^{th} \text{ bin is empty and } \pi_e(S) \cap \Omega_j \neq \emptyset \\ Empty & others \end{cases} \tag{8}$$

$$j = \lfloor \frac{i}{2} \rfloor + (i+1)\%2 \tag{9}$$

where $\pi_e(S)$ $(\pi_o(S))$ denotes the even (odd) hash values of $S$ and $\Omega_i$ denotes $i$-th partition of the range space of $\pi(S)$. The formula for odd bins is similar.

## 3.2 Time and Memory Overhead

For time overhead, AHash keeps comparable runtime efficiency with OPH, the fastest variant of MinHash. Although each bin maintains one more hash values than OPH, the additional runtime cost is negligible because of the memory access locality of the insertion (two values in a bin). Note that AHash has much less empty bins which require densification compared with OOPH, which can save the time cost.

For memory overhead, AHash also provides $k$ MinHash values, which is the same as MinHash and OOPH. During the *Insertion*, the cost of storing two hash values in a bin is acceptable. For the common setting (hundreds of bins and 64-bit MinHash values), the vector of bins with two values per bin can be totally fitted into L1 cache (usually larger than 32 KB), the fastest memory.

## 4 Experiments and Applications

### 4.1 Setup

We use three publicly available datasets [2]:
a) **RCV1:** The dataset is a collection for text categorization. It has $20,242$ sets and the size of the set is on average 73. The dimensionality (range space) of elements is $47,236$.
b) **NEWS20:** The dataset is a collection of newsgroup documents. It has $19,996$ sets and the size of the set is on average 402. The dimensionality (range space) of elements is $1,355,191$.
c) **URL:** The dataset is a collection for identifying suspicious URLs. It has $100,000$ sets and the size of the set is on average 115. The dimensionality (range space) of elements is $3,231,961$.

We use two metrics to measure the performance:
a) **Mean Square Error (MSE)** is defined as $\mathbb{E}[(\hat{J} - J)^2]$. We use MSE to measure the accuracy of computing the *Jaccard Similarity*.
b) **F1-score** is defined as $\frac{2P \cdot R}{(P+R)}$, where $P$ is the precision rate and $R$ is the recall rate. We use $F_1$-*score* to evaluate the performance of fast near neighbour search.

We implemented all algorithms in C++ which are publicly available [3] on GitHub. All experiments are conducted on laptop with 2.9 GHZ Intel Core i7 CPU.

---

[2]https://www.csie.ntu.edu.tw/ cjlin/libsvmtools/datasets/binary.html
[3]https://github.com/AHashCodes/AHash

## 4.2 Accuracy and Speed

Table 1: Pairs of Sets

| Pairs | $JS$ | $|S_1|$ | $|S_2|$ |
|-------|------|---------|---------|
| A | 0.27 | 750 | 567 |
| B | 0.34 | 750 | 748 |
| C | 0.44 | 750 | 870 |
| D | 0.53 | 750 | 751 |
| E | 0.61 | 750 | 892 |
| F | 0.71 | 750 | 719 |

Table 2: Time (Second) to compute 256 MinHash values

| Hash | RCV1 | NEWS20 | URL |
|------|------|--------|-----|
| MinHash | 3.44 | 20.45 | 25.83 |
| OOPH | 0.024 | 0.13 | 0.18 |
| AHash | 0.025 | 0.14 | 0.18 |

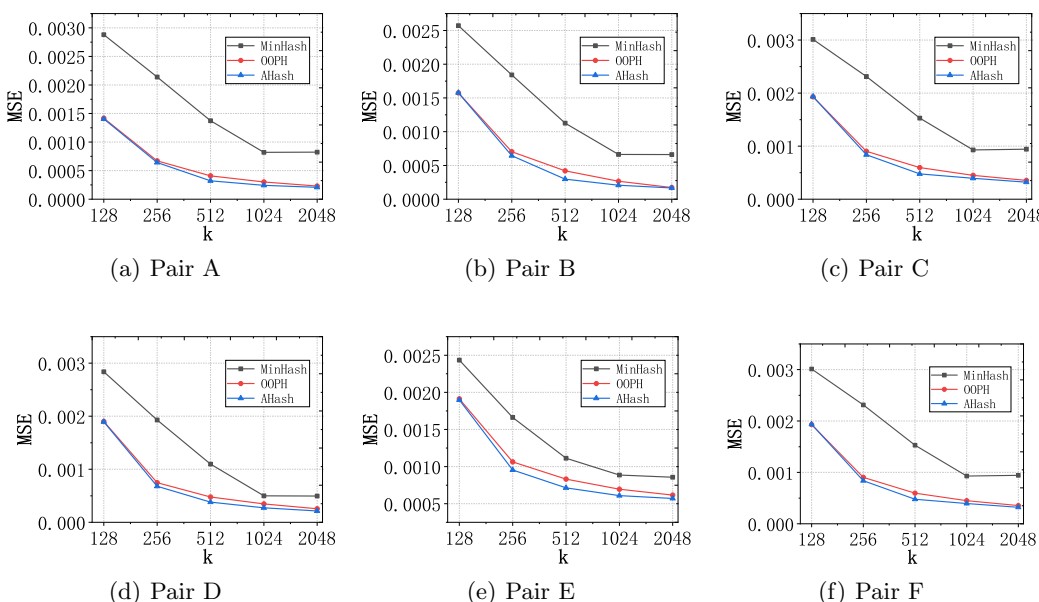

(a) Pair A      (b) Pair B      (c) Pair C

(d) Pair D      (e) Pair E      (f) Pair F

Figure 3: Average MSE in Jaccard Similarity estimation with varing the number of MinHash values ($k$). Results are averaged over 2000 repetitions.

To measure the accuracy of AHash, we generate 6 pairs of sets from the data RCV1, with varying similarities. Because it is difficult to use original sets to form the pairs with high similarity, we unite some original sets as a new set to form the pairs. Detailed statistics is shown in Table 1.

Figure 3 presents the results about the accuracy of computing similarities with varying $k$ (the number of Minhash values). Regardless of the similarity level, AHash is more accurate than OOPH and MinHash (*e.g.*, at $k = 512$, the MSE of AHash is lower than OOPH by 13.2%-31.7%). For Figure 3, the absolute values of MSE are too small, so it's a little hard to see the difference in the values. But the improvements are significant (up to 29.021%), e.g., the MSE of AHash is 29% lower than OOPH with $k = 512$ in Figure 3(b).Note that the gain of AHash is more significant when the number of bins is appropriate (not extremely small or large). The reason is that if $k$ is extremely small, all bins are non-empty. Therefore, AHash has no need for amortization. To the other extreme, if $k$ is extremely large, only a small portion of bins are non-empty and the amortization plays a limited role. Fortunately, with a practical and commonly used configuration, AHash presents significant gains.

Table 2 shows that the speed of computing $k$ MinHash values of AHash is comparable with OOPH, which is two orders of magnitude faster than MinHash. AHash does not hurt the runtime efficiency of OOPH. AHash and OOPH both use much less runtime to more accurately compute the similarity.

### 4.3 Linear Learning

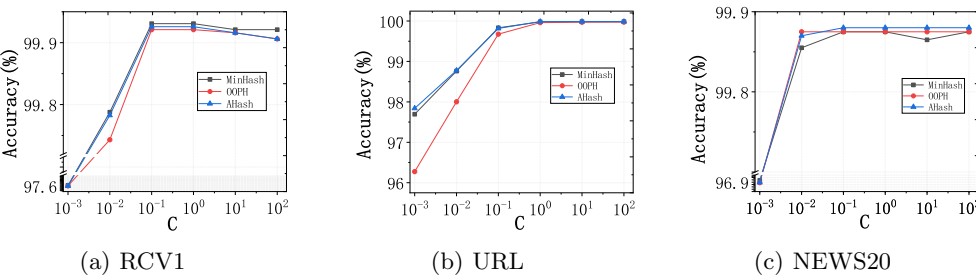

(a) RCV1          (b) URL          (c) NEWS20

Figure 4: SVM test accuracy. We set $k$ as 150 and apply the b-bit hash with $b$ as 8.

We use LIBLINEAR (Fan et al., 2008) to train a $L_2$-*nomarlized* SVM to measure the effectiveness of AHash in reducing the dimensionality of the training data. We experimented with varing penalty parameter $C$, which follows other works (Li et al., 2011; Yu et al., 2012). In this way, it is more easy to reproduce our experiments.

Figure 4 shows that AHash can achieve 99% test accuracies and outperforms OOPH. Compared with training with the original high-dimensional data, the training based on the new feature vectors built by AHash is usually two orders of magnitude faster. Figure 4 shows AHash can further reduce the error, e.g., the error of AHash on average is 46.81% lower than OOPH in Figure 4(b). AHash can achieve the satisfied accuracy, close to 100%.

### 4.4 Near Neighbour Search

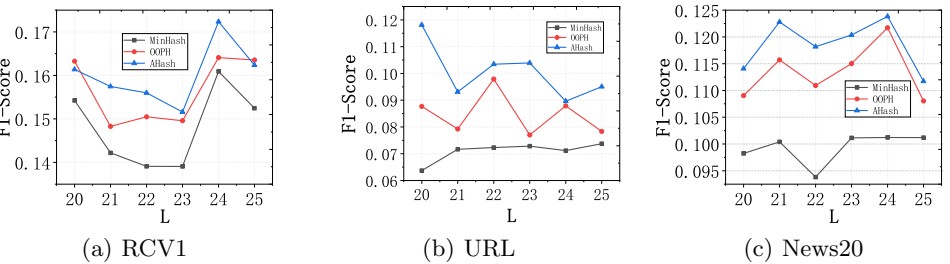

(a) RCV1          (b) URL          (c) News20

Figure 5: $F_1$-score of fast near neighbour search. We set $k$ as 10 and apply the b-bit hash with $b$ as 32. Results are averaged over 2000 queries.

We apply AHash with the optimization of $b$-bit hashing (Li et al., 2010) for fast near neighbour search. Given a query set, threshold of the similarity is 0.5, that is, near neighbour search should return sets whose similarity with the query set is more than 0.5. It is important to balance the precision and recall to measure the effectiveness on near neighbour search. Therefore, we use $F_1$-score as the metric.

Figure 5 shows that the $F_1$-score of AHash is significantly higher than MinHash and OOPH on all three datasets. Note that the trend of $F_1$-score with varying $L$ is not monotonous. With $L$ increasing, the number of reported elements and the number of reported near neighbours both increases, but their growth trends are of randomness. In spite of this, AHash can outperform OOPH and MinHash under most of parameter settings.

## 5 Conclusion

We propose Amortization Hashing which can improve the accuracy of One Permutation Hashing and densification strategies without loss in runtime efficiency. AHash outperforms the state-of-the-art OOPH in similarity estimating, large-scale learning and fast near neighbour searching.

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

# A   THEORETICAL ANALYSIS

**Theorem 1  *Unbiased Estimator***

$$\Pr[h^{AHash}(S_1) = h^{AHash}(S_2)] = \frac{|S_1 \cap S_2|}{|S_1 \cup S_2|} = J(S_1, S_2) \tag{10}$$

$$E(\hat{J}_{AHash}) = J \tag{11}$$

where $\hat{J}_{AHash}$ denotes the estimator given by AHash. Theorem 1 is to show the estimator of AHash is following the LSH property and unbiased.

**Proof A.1**  *For convenience, we define some Boolean values:*

$$I_{emp,i}_{0 \leq i < k} = \begin{cases} 1 & \textit{For both } i^{th} \textit{ bins of } S_1 \textit{ and } S_2, \textit{ no elements falls in them} \\ 0 & \textit{otherwise} \end{cases} \tag{12}$$

$$I_{mat,i}_{0 \leq i < k} = \begin{cases} 1 & \textit{Before Amortization, the values of the } i^{th} \textit{ bins of } S_1 \textit{ and } S_2 \textit{ are the same} \\ 0 & \textit{otherwise} \end{cases} \tag{13}$$

$$I^A_{emp,i}_{0 \leq i < k} = \begin{cases} 1 & h_i^{AHash}(S_1) = h_i^{AHash}(S_2) = Empty \\ 0 & otherwise \end{cases} \tag{14}$$

$$I^A_{mat,i}_{0 \leq i < k} = \begin{cases} 1 & h_i^{AHash}(S_1) = h_i^{AHash}(S_2) \\ 0 & otherwise \end{cases} \tag{15}$$

*Obviously, for simultaneously non-empty bins, AHash is following the LSH property:*

$$E(I_{mat,i} | I_{emp,i} = 0) = J \tag{16}$$

*Next, we prove that the Amortization for simultaneously empty bins is unbiased:*

$$I_{emp,i} = 1 \textit{ and } I^A_{emp,i} = 0 \longrightarrow I_{emp,j} = 1 \qquad (j = \lfloor \frac{i}{2} \rfloor + (i+1)\%2)$$

$$E(I^A_{mat,i} | I_{emp,i} = 1 \textit{ and } I^A_{emp,i} = 0) = E(I_{mat,j} | I_{emp,j} = 1) = J \tag{17}$$

*If $I^A_{emp,i} = 1$, i.e., there is still the pair of bins which are simultaneously empty after Amortization, we use unbiased Optimal Densification (Shrivastava, 2017) to fill such bins.*

**Theorem 2  *Reducing Empty Bins***

$$E(N_{emp}^{AHash}) \leq E(N_{emp}^{OPH}) \tag{18}$$

where $N_{emp} = \sum_{i=1}^{k} I_{emp}^i$ is the number of simultaneously empty bins.

**Proof A.2**  *(Li et al., 2012) proves the expectation of the number of simultaneously empty bins of OPH is:*

$$E(N_{emp}^{OPH}) = y(k) = k \prod_{i=0}^{f} \frac{D(1 - \frac{1}{k}) - i}{D - i} \tag{19}$$

*where $f$ is $|S_1 \cup S_2|$ and $D$ is the dimensionality of the universal hashing function used. The process of OPH can be viewed as randomly throwing $f$ balls into $k$ bins. For AHash, because $k$ bins are paired up, the process of AHash can be viewed as randomly throwing $f$ balls into $\frac{k}{2}$ bins. Obviously, $N_{emp}$ of AHash is smaller than OPH:*

$$E(N_{emp}^{AHash}) = y(\frac{k}{2}) \leq y(k) \tag{20}$$

**Theorem 3  *Improving Variance***

$$Var(\hat{J}_{AHash}) \leq Var(\hat{J}_{OOPH}) \tag{21}$$

**Proof A.3** *We define two Boolean values with $1\{x\}$ being the indicator function that takes 1 when $x$ is true and otherwise 0.*

$$M_j^A = 1\{I_{emp,j}^A = 0 \text{ and } h_j^{AHash}(S_1) = h_j^{AHash}(S_2)\} \tag{22}$$

$$M_j^E = 1\{I_{emp,j}^A = 1 \text{ and } h_j^{AHash}(S_1) = h_j^{AHash}(S_2)\} \tag{23}$$

*The estimator given by AHash and the variance of AHash can be rewritten as follows using these two values.*

$$\hat{J}_{AHash} = \frac{1}{k} \sum_{j=1}^{k} [M_j^A + M_j^E] \tag{24}$$

$$Var(\hat{J}_{AHash}) = E((\frac{1}{k} \sum_{j=1}^{k} [M_j^A + M_j^E])^2) - (E(\hat{J}_{AHash}))^2 \tag{25}$$

*Theorem 1 proves $(E(\hat{J}_{AHash}))^2 = J^2$. Define two notations:*

$$N_{emp}^{OPH} = \sum_{i=1}^{k} I_{emp,i} \qquad N_{emp}^{AHash} = \sum_{i=1}^{k} I_{emp,i}^A \tag{26}$$

*OOPH uses conditional expectation to simplify variance analysis. To conveniently compare with OOPH, we follow its method where $E(.|m)$ means $E(.|k - N_{emp}^{AHash} = m)$. We compute $f(m) = E((\frac{1}{k} \sum_{j=0}^{k-1} [M_j^E + M_j^N])^2|m)$ By expanding,*

$$k^2 f(m) = E[\sum_{i=1}^{k} ((M_i^A)^2 + (M_i^E)^2)|m] + E(\sum_{i \neq j} M_i^A M_j^A|m) \\ + E(\sum_{i \neq j} M_i^A M_j^E|m) + E(\sum_{i \neq j} M_i^E M_j^E|m) \tag{27}$$

*Because $M_i^A$ and $M_i^E$ are Boolean values,*

$$E[\sum_{i=1}^{k} ((M_i^A)^2 + (M_i^E)^2)|m] = E[\sum_{i=1}^{k} (M_i^A + M_i^E)|m] = k \times J \tag{28}$$

*To analyze another three terms, we provide four probability equations for bins before the amortization.*

$$Pr(I_{emp,j} = 1) = P_e = (\frac{k-1}{k})^{|S_1 \cup S_2|} \tag{29}$$

$$Pr(I_{emp,j} = 0) = P_f = 1 - P_e \tag{30}$$

*$Pr(I_{emp,j}=0$ and the mins from two j-th bins have one and only one position)*

$$= P_s = P_f \times 2 \times (\frac{1}{2})^{|S_1 \cup S_2|/(k-N_{emp}^{OPH})} \tag{31}$$

*$Pr(I_{emp,j}=0$ and the mins from two j-th bins have two positions (odd/even))*

$$= P_d = P_f - P_s \tag{32}$$

*Define one notation:*

$$\hat{J} = \frac{|S_1 \cap S_2| - 1}{|S_1 \cup S_2| - 1} \tag{33}$$

*The values of following three terms is proved by classifying different combinations of bins and applying probability equations and the property of indicator functions.*

$$E(\sum_{i \neq j} M_i^A M_j^A|m) = m(m-1)[J\hat{J}\frac{k-2}{k-1} + J\frac{1}{k-1} \times P_s P_e + J\hat{J}\frac{1}{k-1} \times 2P_d P_e] \leq m(m-1)J\hat{J} \tag{34}$$

$$E(\sum_{i \neq j} M_i^A M_j^E | m) = 2m(k-m)[\frac{J}{m} + \frac{(m-2)J\hat{J}}{m} + J\frac{1}{m} \times P_s P_e + J\hat{J}\frac{1}{m} P_d P_e]$$
$$\leq 2m(k-m)[\frac{J}{m} + \frac{(m-1)J\hat{J}}{m}] \tag{35}$$

$$E(\sum_{i \neq j} M_i^E M_j^E | m) = (k-m)(k-m-1)[\frac{J}{m} + \frac{(m-1-\frac{m-1}{k-1})J\hat{J}}{m} + J\hat{J}\frac{1}{k-1} \times 2P_d P_e]$$
$$+J\frac{1}{k-1} \times P_s P_e + J\hat{J}\frac{1}{k-1} \times P_f P_f] \leq (k-m)(k-m-1)[\frac{J}{m} + \frac{(m-1)J\hat{J}}{m}] \tag{36}$$

*Hence, we get that*

$$k^2 f(m) \leq h(m) = k \times J + m(m-1)J\hat{J} + 2m(k-m)[\frac{J}{m} + \frac{(m-1)J\hat{J}}{m}]$$
$$+(k-m)(k-m-1)[\frac{J}{m} + \frac{(m-1)J\hat{J}}{m}] \tag{37}$$

*For OOPH, the variance is*

$$Var(\hat{J}_{OOPH}) = E((\frac{1}{k}\sum_{j=0}^{k-1}[M_j^N + M_j^D])^2) - (E(\hat{J}_{OOPH}))^2, \tag{38}$$

$$M_j^N = 1\{I_{emp,j} = 0 \ and \ h_j^{OPH}(S_1) = h_j^{OPH}(S_2)\}$$
$$M_j^D = 1\{I_{emp,j} = 1 \ and \ h_j^{OOPH}(S_1) = h_j^{OOPH}(S_2)\}. \tag{39}$$

*For OOPH, the conditional expectation is*

$$E((\frac{1}{k}\sum_{j=1}^{k}[M_j^N + M_j^D])^2 | k - N_{emp}^{OPH} = m') = h(m') \tag{40}$$

$$\frac{\partial h(m)}{m} = \frac{k^2 - k + m^2}{m^2}(J\hat{J} - J) \leq 0 \ and \ m \geq m'$$
$$\implies h(m) \leq h(m') \implies Var(\hat{J}_{AHash}) \leq Var(\hat{J}_{OOPH}) \tag{41}$$

Because AHash reduces $N_{emp}$, AHash using Optimal Densification (Shrivastava, 2017) can achieve lower variance compared with OOPH.

