# OpenReview forum: "AHash: A Load-Balanced One Permutation Hash"
_ICLR.cc/2020/Conference — Reject_

### Official Review · AnonReviewer2 · 2019-10-23
**Official Blind Review #2**

**Rating:** 1

**Review:**

*** Summary ***
MinHash is a well-known method for approximating set similarities in terms of the jaccard similarity. The idea is to use k random permutations (hashes) on all elements of the sets and check how often two sets hash into the same bucket. Larger values of k yield more accurate estimates of the set similarities but require more time to compute. One permutation hashing (OPH) aims to reduce the number of hash computations per element to 1 and maintaining bins. However, some of those bins may remain empty which negatively influences the similarity estimate. Optimal OPH (OOPH) hashes empty bins to non-empty bins and effectively reuses bins. This paper proposes Amortization Hashing (AHash) which reduces the occurrence of empty bins and thus leads to better similarity estimates.

*** Evaluation ***
The paper proposes an interesting idea for approximating set similarities much faster than MinHash. However, I have some issues with the submission.

I believe that the manuscript has a limited impact. The approach performs on par or marginally better as OOPH within the first and only reasonable experiment (4.2). As the authors state themselves on the page break 9/10, the advantage of AHash vanishes for small and large values of k. Hence, AHash only benefits of moderate choices of k. Moreover, I can see that OOPH might have a minor problem of estimating the set similarities which AHash aims to fix, but why should it outperform MinHash in terms of accuracy? Why are the pairs of set only chosen from RCV1? Why those particular set sizes? Why does no plot show standard deviation/error?

The remaining experiments yield very limited insight. Considering a linear SVM on standard datasets where the test error is >99.8% seems to be obsolete. In addition, the most important parameter k is held fix to an arbitrary value. Same holds for b. Since AHash only benefits from moderate sizes of k, why was k chosen in favor of AHash? The performance should definitely be shown in dependence of k. Instead, the most unimportant parameter (C) is varied. This should have been done in a proper cross-validation. Similar arguments hold for the near neighborhood search. What is the query set being used?

There are more flaws within the manuscript. The mathematical presentation is rather poor. The theorems lack text and assumptions and solely consist of equations. The corresponding proofs are also short on text and hard to follow. Unfortunately, there is no analysis of the expected error as a function of k. The proof of Theorem 3 is almost two pages and should be moved to the supplementary material since it does not provide much insight; it just distracts the reading flow. In addition, every equation is numbered but none is ever referenced. The citation style (numbers in round brackets) is really uncommon and can be easily confused with equation numbers. Most importantly, I want to note that a different font was used and that the spacing was clearly tricked in several places (e.g. within Section 4). This makes it especially hard to judge whether the manuscript has the correct length.


*** Further Comments ***
- The font was changed. It does not match the font of the other submissions.
- The spacing is tricked in several places, especially in Section 4.
- Links [1,29] should not be references but footnotes.
- Citations should never be in round brackets like (1), because they can be confused with equation numbers. Instead they should be in square brackets like [1] or, more preferably, the natbib package should be used as in the ICLR style guidelines.
- What does OOPH stand for? It is never stated.
- Every equation has a number, but none is ever referenced.
- Math/Equations are part of the text and should be treated as such, i.e., there should be proper punctuation marks.
- What is a 2-universal hashing?
- Algorithm 1: "output range" sounds like an interval whereas the number of distinct hash values is meant.
- "(14) proposed", no past tense
- Instead of "(11) proposes", please use "Shrivastava and Li [11] propose"
- Why "Theorem 1" and "Proof 3.1"?
- Why are the theorems lacking the assumptions and text? They basically consist of equations.
- Theorem 3 should have a "less or equal" instead of a "strictly less".
- Eq. (40): "0andm"
- Why does Proof 3.3 have a end of proof sign (not right-aligned) but the other proofs don't?
- None of the experimental results shows standard deviations/errors although the experiments are repeated several times. Why? It would be also nice to see whether the approximation tends to over- or underestimate J. This could be done with a violin plot.
- How are the pairs of sets in Section 4.2 chosen and why only from RCV1? This seems to be the most important experiment.
- Why is k (and b) fixed to an arbitrary value in the remaining experiments? Please select C within a proper cross-validation.
- There are a lot of enumerations which unnecessarily make the manuscript longer, e.g. in Sections 1.4, 2.1 and 4.1. In addition, the proof of Theorem 3 almost takes two pages but is not super informative. It should be moved to the supplementary material. This in combination with the font mismatch makes it difficult to determine the real length of the submission.
- It is nice that the source code is published online, but uncommented c++ code is not really helpful.


**Experience Assessment:**

I have read many papers in this area.

**Review Assessment: Checking Correctness Of Derivations And Theory:**

I assessed the sensibility of the derivations and theory.

**Review Assessment: Checking Correctness Of Experiments:**

I assessed the sensibility of the experiments.

**Review Assessment: Thoroughness In Paper Reading:**

I read the paper thoroughly.

---

### Official Review · AnonReviewer1 · 2019-11-02
**Official Blind Review #1**

**Rating:** 6

**Review:**

This paper proposes a new hashing method based on MinHash. It aims to improve the computational complexity of the original MinHash and reduce the error of One Permutation Hashing. The proposed method is mainly assigning bins into paired bins, odd bins and even bins, and obtain two hash values, an even min and an odd min, for each bin. It finally reassigns nearby hash values to empty bins. The paper provides theoretical analyses that show the proposed method gives 1) unbiased estimator; 2) fewer empty bins 3) smaller variance. Finally the paper did some experiments on several applications including SVM and nearest neighbor search. The results show improvement over MinHash and OOPH.

The paper is mostly well written and easy to read. I have a few questions.

1. My main concern is the lack of proof details. I tried to read through Proof 3.1. However, I found it is hard to follow. I'd suggest to add more details and explaination in the proof. For example, how is Eq. (17) obtained?

2. The floor symbol \lfloor, \rfloor in Eq. (9) seems to be misused. It seems that the paper wants  $\lfloor n/2 \rfloor$ to return the largest even integer that is less than or equal to the given value, n. Right?

Overall, I vote for weak accept as the proposed is novel and the theoretical results seem reasonable to me.

**Experience Assessment:**

I do not know much about this area.

**Review Assessment: Checking Correctness Of Derivations And Theory:**

I assessed the sensibility of the derivations and theory.

**Review Assessment: Checking Correctness Of Experiments:**

I assessed the sensibility of the experiments.

**Review Assessment: Thoroughness In Paper Reading:**

I read the paper at least twice and used my best judgement in assessing the paper.

---

### Official Review · AnonReviewer3 · 2019-11-02
**Official Blind Review #3**

**Rating:** 3

**Review:**

The paper proposes a bin split strategy for one permutation hashing which reduces the expected number of empty bins. The idea is to use two sets of hashes (odd and even) to make more use of the data. The strategy reduces the variance of existing densification scheme. Experiments show that the proposed method gives lower MSE, good classification accuracy and better NN search performance.

Overall, the idea is interesting, although the improvements will only occur in very much corner cases. The wiring can be improved. The author also use ( ) instead of [] for citations which made the paper difficult to read (among with other writing problems). The theory (and its correctness) is difficult to be justified. The authors are suggested to use plots to verify the theoretical results.



**Experience Assessment:**

I have read many papers in this area.

**Review Assessment: Checking Correctness Of Derivations And Theory:**

I assessed the sensibility of the derivations and theory.

**Review Assessment: Checking Correctness Of Experiments:**

I assessed the sensibility of the experiments.

**Review Assessment: Thoroughness In Paper Reading:**

I read the paper at least twice and used my best judgement in assessing the paper.

---

### Official Review · AnonReviewer4 · 2019-11-05
**Official Blind Review #4**

**Rating:** 6

**Review:**

In this paper, the authors proposed an efficient minwise hashing method, namely Amortization Hashing (AHash), for solving the issue of the unbalanced loading during hash code generation. Specifically, in AHash, two steps, called Insertion and Amortization, are designed to balance the load in order to reduce the empty bins. In addition, a detailed theoretical analysis was provided. The authors evaluated the proposed AHash on three text categorization benchmarks, i.e., RCV1, NEWS20, and URL, comparing against the previous min-hashing algorithms including OOPH and MinHash. The experimental results show the effectiveness of the proposed hashing method.

Some comments on this work are listed as follows.
1. This paper is well motivated and structured, which may serve as a guidance for other works.
2. Three tasks, similarity estimating, large-scale learning, and fast near neighbor searching reduction, are adopted to validate the proposed AHash. The experimental results show that the proposed method can achieve state-of-the-art performance.
3. It seems that the effectiveness of the proposed method is OK but not surprising. The improvement of AHash over OOPH is unobtrusive, given the results shown in Table 2, Fig. 3, and Fig. 4.
4. The authors claimed that more signiﬁcant performance can be achieved when the number of bins is appropriate on the task of computing similarities. Is there any theoretical guarantee for such an appropriate condition?
5. The proposed method still cannot entirely solve the problem of empty bins, where a densifying operation may also be needed.
Minor error:
A typo exists in Eq. (10), where one “)” on the left-hand side is needless.

Overall, the proposed AHash in this work is interesting and seems effective when dealing with the problem of the unbalanced load. However, there still exist quite a few issues in the current version, which need more explanations and clarifications.


**Experience Assessment:**

I have published in this field for several years.

**Review Assessment: Checking Correctness Of Derivations And Theory:**

I carefully checked the derivations and theory.

**Review Assessment: Checking Correctness Of Experiments:**

I carefully checked the experiments.

**Review Assessment: Thoroughness In Paper Reading:**

I read the paper at least twice and used my best judgement in assessing the paper.

---

### Decision · Program_Chairs · 2019-12-19

**Decision:**

Reject

**Comment:**

This paper proposes a load-balanced hashing called AHash that balances the load of hashing bins to avoid empty bins that appear in some minwise hashing methods.
Reviewers found the work interesting and well-motivated. Authors addressed some clarity issues in their rebuttal. However the impact appeared quite limited, and the experimental validation limited to few realistic experiments that did not alleviate this concern.
We thus recommend rejection.